

# Translation and validation of the Bahasa Malaysia version of the Nasal Obstruction Symptom Evaluation scale (M-NOSE)

Addina Mohd Baki[1], Suzina Sheikh Ab Hamid[1], Rosli Mohd Noor[2] and Baharudin Abdullah[1]

[1] Department of Otorhinolaryngology-Head and Neck Surgery, School of Medical Sciences, Universiti Sains Malaysia, Kubang Kerian, Kelantan, Malaysia

[2] Department of Otorhinolaryngology-Head and Neck Surgery, Hospital Raja Perempuan Zainab 2, Jalan Hospital, Kota Bharu, Kelantan, Malaysia

## ABSTRACT

**Background**. The Nasal Obstruction Symptom Evaluation (NOSE) is a questionnaire to assess patients with nasal obstruction. The aim of this study was to translate and validate NOSE to the Malay version NOSE (M-NOSE).

**Methods**. The NOSE questionnaire was translated to Malay language and back to English. Psychometric properties consisting of reliability, reproducibility, validity, responsiveness were appraised in patients with nasal obstruction due to deviated nasal septum and healthy asymptomatic controls.

**Results**. A total of 126 participants were recruited. There was significant difference between patients and controls for all items and the total score (all $p < 0.001$). The correlation was moderate to strong between all items and total score ($r = 0.71$ to $0.8$) and fair to moderate for the inter-items correlations ($r = 0.31$ to $0.70$). Internal consistency for M-NOSE was good ($\alpha = 0.81$). The test–retest for each item demonstrated no significant difference. There was significant difference of the pre- and post-operative mean for each item and total score (all $p < 0.001$) with good response sensitivity (effect size, $d = 4.91$).

**Conclusions**. The M-NOSE has satisfactory reliability, internal consistency, reproducibility and responsiveness. It is a valid and convenient tool in the assessment of the impact and treatment outcome of nasal obstruction.

Corresponding author
Baharudin Abdullah,
profbaha@gmail.com

## INTRODUCTION

Blocked nose or nasal obstruction is a sensation of discomfort when there is a feeling of restricted or insufficient airflow going through the nose (*Esmaili & Acharya, 2017*). It is a condition that greatly affects a patient's quality of life (QOL) and general wellbeing (*Valero et al., 2018*). Diagnostic evaluation includes nasoendoscopic examination, rhinomanometry, acoustic rhinometry, and computed tomography (*Aziz et al., 2014*). However, the evaluation may not reflect the level of severity that the patient perceives.

Health-related QOL questionnaires are one of the tools for the assessment of the impact of symptoms on the quality of life and treatment outcomes including the effectiveness of surgery (*Sitlinger & Zafar, 2018*).

The Nasal Obstruction Symptom Evaluation (NOSE) is a health-related QOL questionnaire for the assessment of patients with nasal obstruction (*Stewart et al., 2004*). The NOSE questionnaire has a good internal consistency and reliability, making it a useful tool to assess the impact of nasal obstruction on the patient's QOL (*Stewart et al., 2004*). The questionnaire is composed of five obstruction-related items, which evaluate the severity of symptoms that the patient experienced over the past month. All items are scored using a five points Likert scale. It has been successfully adapted to other languages with a validity and reliability similar to the original version. The French, Greek and Turkish versions of NOSE were validated with satisfactory internal consistency, reliability, reproducibility, and responsiveness similar to the original English version (*Marro et al., 2011*; *Lachanas et al., 2014*; *Onerci Celebi et al., 2018*).

There is limited availability of a validated NOSE in the Asian language. The use of English NOSE may not be reflective of the actual symptoms' severity and not possible in those who are not well versed with the language. Malay language is widely used in the Malay Archipelago which encompasses the countries of Brunei Darussalam, Indonesia, Malaysia, the Philippines, and Singapore (*United Nations, 2019*; *Sodhi et al., 2006*). To the best of our knowledge no Malay version of the NOSE questionnaire currently exists. The aim of this study was to translate the NOSE into Malay Language (M-NOSE) and evaluate its validity and reliability.

## MATERIALS AND METHODS

### Study design

A prospective questionnaire validation study was conducted at two tertiary hospitals from May 2020 until July 2021. The study was approved by the Human Research Ethics Committee, Universiti Sains Malaysia (USM) (No: USM/JEPeM/20010021). Permission from the original author of NOSE questionnaire was obtained (*Stewart et al., 2004*). Written consent was obtained from all participants and was performed in adherence with the Declaration of Helsinki.

### Subjects

Two groups of patients were recruited. The first group included patients who had nasal obstruction for more than 3 months due to deviated nasal septum. Patients over 18 years who were candidates for septoplasty were included. Patients with nasal pathologies such as tumour, granulomatous disease, facial trauma, nasal valve collapse, craniofacial abnormality, history of radiotherapy, previous septal surgery and nasal allergy or rhinosinusitis were excluded. The history, nasal examination, and skin prick test were used to rule out nasal allergies. The second group consisted of healthy asymptomatic controls aged above 18 years old without history of nasal obstruction and/or use of intranasal medication and who were not candidates for nasal surgical procedure. All participants were able to read and write Malay language.

## Translation of M-NOSE

Two forward translations of the original English NOSE to Malay language were made independently by a bilingual professional translator without medical background from the language center and a bilingual junior medical officer from the department of otorhinolaryngology. Two independent otolaryngologists further refined the questionnaire and following discussion, a consensual Malay version was established. Backward translation of the Malay version was performed by two independent bilingual professional translators without medical background to ensure the translated version has the same content as the original. The final version of M-NOSE was produced by an expert committee comprising of seven otorhinolaryngologists, one methodologist, the translators and language professionals (Table S1). Content validity was approved by the expert committee.

## Validation of M-NOSE

All participants in the intervention group were required to answer the M-NOSE thrice. Initially, patients were required to complete the M-NOSE on the same day during clinic visit. Subsequently, they were given 2 weeks' appointment from the first visit. The final session was 3 months after surgery. The control group completed the questionnaire once. To avoid any external influence, all participants were informed to answer the questions alone. In addition, a 10 cm horizontal visual analog scale (VAS) measuring the subjective sensation of nasal obstruction was included with the M-NOSE during enrollment. The construct validity was determined by correlating the responses obtained for each item with the other items and then with the overall score. The criterion validity was assessed by comparing the M-NOSE item scores to 10 cm Visual Analog Scale, with score ranging from 0 (very bad) to 10 (very good). The M-NOSE scores of patients and controls were compared to evaluate the discriminant validity. The psychometric properties were assessed by test-retest procedure, internal consistency, correlation intra- and inter-scores, and response sensitivity. Test–retest reliability was conducted 2 weeks after the first test for the patient group. Patients with change of symptoms due to upper respiratory tract infection during the period between the test-retest questionnaire were excluded from the study. Three months after surgery, patients were given the M-NOSE to measure their response.

## Statistical analysis

Data were entered and analyzed *via* IBM SPSS version 26 (IBM Corp, Armonk, NY, USA). The numerical variables were expressed as means, and the categorical variables were expressed as frequencies. The socio-demographic characteristics among patient and control groups were analyzed by $t$-test and chi square test. The discriminant validity was evaluated by $t$-test or Mann–Whitney test. The construct validity between items and total items was assessed using Spearman correlation and Pearson correlation tests. The M-NOSE scores were correlated with the VAS score to determine the criterion validity. Reliability or internal consistency of items in the scale was tested and interpreted as fair for Cronbach's alpha between 0.70 and 0.79, good between 0.80 and 0.89, and excellent above 0.90 (*Heale & Twycross, 2015*). Test–retest reliability was assessed and the score of each item with other item and total score was correlated. A correlation coefficient more than 0.3

**Table 1  Characteristics between control and intervention groups.**

| Variable | Control ($n = 63$) | Intervention ($n = 63$) | Mean difference (95% CI) | $t$-statistic (df) | $p$-value |
|---|---|---|---|---|---|
| Age (years), mean (SD) | 35.92 (6.98) | 33.94 (11.48) | 1.98 (−1.37, 5.34) | 1.17 (124) | 0.243[a] |
| Gender, $n$ (%) | | | | | |
| Female | 31 (49.2) | 22 (34.9) | – | 2.64 (1) | 0.104[b] |
| Male | 32 (50.8) | 41 (65.1) | | | |

Notes.
[a] $t$-test.
[b] chi-square test.

considered acceptable for inter-item correlation and value more than 0.50 acceptable for item-total score correlation (*DeVon et al., 2007*). The strength of all correlations was graded as follows: less than 0.3 (poor), 0.3 to 0.5 (fair), between 0.5 to 0.8 (moderate), above 0.8 (strong) (*Chan, 2003*). The responsiveness was assessed by comparing the pre-operative and post-operative M-NOSE scores using Wilcoxon-signed rank test and paired $t$-test. Additionally, the response sensitivity was assessed by determining the Cohen's effect size ($d$) (*Sullivan & Feinn, 2012*). Effect size between 0.2 and 0.5 was interpreted as small effect; between 0.5 and 0.8 medium effect; between 0.8 and 1.0 large effect; and greater than 1.0 very large effect (*Leech, Barrett & Morgan, 2005*).

# RESULTS

## Participants characteristics
A total of 126 participants (63 patients and 63 controls) were recruited. The characteristics of patients and control group are shown in Table 1. No significant difference of age ($p = 0.243$) and gender ($p = 0.104$) was seen between the two groups.

## Discriminant validity
There was a significant difference of the mean score between intervention and control groups for all items and the total score (all $p < 0.001$). The intervention group had higher M-NOSE score compared to control for all items (Table 2).

## Construct validity
The values of inter-item and between-item correlation coefficients and the total score are shown in Table 3. The correlations were moderate to strong between all items and total score ($r = 0.71$ to 0.8) while the inter-items correlations were fair to moderate ($r = 0.31$ to 0.70).

## Criterion validity
The VAS score has a significant correlation with all items in the M-NOSE ($p < 0.001$) (Table 4). The correlation coefficient ranged from 0.81 to 0.86, indicating that the M-NOSE items have a strong correlation with VAS score. In addition, there was a strong correlation between VAS score and M-NOSE total score ($r = 0.86$, $p < 0.001$).

**Table 2  Difference of M-NOSE between control and intervention groups.**

| Item | Median (interquartile range) | | z-statistic | p-value[a] |
|---|---|---|---|---|
| | Control (n = 63) | Intervention (n = 63) | | |
| Nasal congestion | 0.00 (0.00) | 3.00 (1.00) | −9.61 | <0.001 |
| Nasal blockage | 0.00 (0.00) | 3.00 (1.00) | −10.14 | <0.001 |
| Trouble breathing through nose | 0.00 (0.00) | 3.00 (0.00) | −10.27 | <0.001 |
| Trouble sleeping | 0.00 (0.00) | 3.00 (1.00) | −9.98 | <0.001 |
| Unable to get air through my nose during exercise or exertion | 0.00 (0.00) | 3.00 (2.00) | −9.96 | <0.001 |
| Total score | 0.00 (2.00) | 15.00 (5.00) | −9.82 | <0.001 |
| Total score x 5 | 0.00 (10.00) | 75.00 (20.00) | −9.82 | <0.001 |

Notes.
[a] Mann–Whitney test.

**Table 3  Inter-item and item-total correlation of M-NOSE.**

| Correlation | | Correlation coefficient[a] | p-value[b] |
|---|---|---|---|
| Nasal congestion | Nasal blockage | 0.64 | <0.001 |
| | Trouble breathing through nose | 0.49 | <0.001 |
| | Trouble sleeping | 0.43 | <0.001 |
| | Unable to get air through my nose during exercise or exertion | 0.39 | 0.002 |
| | Total score | 0.75 | <0.001 |
| Nasal blockage | Trouble breathing through nose | 0.62 | <0.001 |
| | Trouble sleeping | 0.40 | 0.001 |
| | Unable to get air through my nose during exercise or exertion | 0.31 | 0.014 |
| | Total score | 0.71 | <0.001 |
| Trouble breathing through nose | Trouble sleeping | 0.56 | <0.001 |
| | Unable to get air through my nose during exercise or exertion | 0.51 | <0.001 |
| | Total score | 0.76 | <0.001 |
| Trouble sleeping | Unable to get air through my nose during exercise or exertion | 0.70 | <0.001 |
| | Total score | 0.82 | <0.001 |
| Unable to get air through my nose during exercise or exertion | Total score | 0.79 | <0.001 |

Notes.
[a] Spearman's correlation.
[b] p significant at the <0.01 level (2-tailed).

## Internal consistency

Cronbach's alpha of M-NOSE was 0.81 indicating a good reliability. With the exception of 'nasal congestion', the internal consistency would be reduced if any items from the scale were deleted (Table 5).

**Table 4  Correlation between visual analogue scale score and M-NOSE items and total score.**

| Correlation | | Correlation coefficient[a] | p-value |
|---|---|---|---|
| Visual analogue scale score | Nasal congestion | 0.81 | <0.001 |
| | Nasal blockage | 0.86 | <0.001 |
| | Trouble breathing through nose | 0.86 | <0.001 |
| | Trouble sleeping | 0.82 | <0.001 |
| | Unable to get air through my nose during exercise or exertion | 0.85 | <0.001 |
| | Total score | 0.86 | <0.001 |

**Notes.**
[a] Spearman's correlation.

**Table 5  Internal consistency of M-NOSE.**

| Item | Cronbach's alpha if item deleted | Cronbach's alpha ($\alpha$) |
|---|---|---|
| Nasal congestion | 0.81 | |
| Nasal blockage | 0.79 | |
| Trouble breathing through nose | 0.79 | 0.81 |
| Trouble sleeping | 0.76 | |
| Unable to get air through my nose during exercise or exertion | 0.77 | |

**Table 6  Difference between first and second tests of M-NOSE.**

| Item | Median (IQR) | | z-statistic | p-value[a] |
|---|---|---|---|---|
| | First (n = 63) | Second (n = 63) | | |
| Nasal congestion | 3.00 (1.00) | 3.00 (0.00) | −1.23 | 0.221 |
| Nasal blockage | 3.00 (1.00) | 3.00 (1.00) | −1.41 | 0.157 |
| Trouble breathing through nose | 3.00 (0.00) | 3.00 (1.00) | −1.39 | 0.166 |
| Trouble sleeping | 3.00 (1.00) | 3.00 (1.00) | −1.73 | 0.083 |
| Unable to get air through my nose during exercise or exertion | 3.00 (2.00) | 3.00 (1.00) | −1.73 | 0.083 |
| Total score | 15.00 (5.00) | 15.00 (3.00) | −1.46 | 0.143 |

**Notes.**
[a] Wilcoxon signed-rank test.

## Test–retest reliability

There was no significant difference between the first and second tests of M-NOSE ($p > 0.05$) (Table 6). There was significant correlation of the items between first and second tests ($p < 0.001$) with correlation coefficient ranging from 0.77 to 0.96 (Table 7).

**Table 7** Correlation between first and second tests of M-NOSE.

| Item | Correlation coefficient | p-value |
|---|---|---|
| Nasal congestion | 0.85[a] | <0.001 |
| Nasal blockage | 0.95[a] | <0.001 |
| Trouble breathing through nose | 0.77[a] | <0.001 |
| Trouble sleeping | 0.96[a] | <0.001 |
| Unable to get air through my nose during exercise or exertion | 0.87[a] | <0.001 |
| Total score | 0.92[b] | <0.001 |

Notes.
[a] Spearman's correlation.
[b] Pearson's correlation.

**Table 8** Difference of M-NOSE between pre- and post-operative groups.

| Item | Mean (SD) | | Mean difference (95% CI) | t-statistic (df) | p-value |
|---|---|---|---|---|---|
| | Pre-operative (n = 63) | Post-operative (n = 63) | | | |
| Nasal congestion | 3.00 (1.00)[a] | 0.00 (0.00)[a] | | −6.89 | <0.001[b] |
| Nasal blockage | 3.00 (1.00)[a] | 0.00 (1.00)[a] | | −7.02 | <0.001[b] |
| Trouble breathing through nose | 3.00 (0.00)[a] | 0.00 (0.00)[a] | | −7.07 | <0.001[b] |
| Trouble sleeping | 3.00 (1.00)[a] | 0.00 (0.00)[a] | | −7.00 | <0.001[b] |
| Unable to get air through my nose during exercise or exertion | 3.00 (2.00)[a] | 0.00 (0.00)[a] | | −6.99 | <0.001[b] |
| Total score | 14.73 (3.00) | 1.11 (1.57) | 13.62 (12.91, 14.33) | 38.34 (62) | <0.001[c] |
| Total score x 5 | 73.73 (14.67) | 5.56 (7.84) | 68.18 (64.68, 71.67) | 38.95 (62) | <0.001[c] |

Notes.
[a] Median (IQR).
[b] Wilcoxon signed-rank test (non-parametric data).
[c] Paired t-test (parametric data).

## Responsiveness (pre- and post-operative evaluation) and response sensitivity

There was a significant difference in the M-NOSE score between pre-operative and post-operative groups ($p < 0.001$) (Table 8). The effect size ($d$) was 4.91, signifying a very large effect in the improvement following surgery.

## DISCUSSION

Besides being used as research instruments, QOL questionnaires are essential for clinicians to evaluate the impact of a particular condition or symptom affecting their patients. Consequently, such measurement tool must possess good reliability and validity to reflect the magnitude of the condition that it evaluates. Whilst a gold standard questionnaire for nasal obstruction does not currently exist, NOSE and VAS are commonly used (*Rhee et al., 2014*). NOSE is a superior tool than VAS as the detailed five items related to nasal

obstruction were incorporated to measure the symptom severity. Furthermore, the NOSE is expedient to measure the impairment in quality of life due to nasal obstruction (*Shukla, Nemade & Shinde, 2020*). Apart from differentiating patients with and without nasal obstruction, NOSE is able to classify the severity into mild, moderate and severe which allows it to be correlated or compared with other clinical assessment (*Lipan & Most, 2013*). By using subjective and objective clinical assessment tools, valuable and reliable information can be obtained to customize therapy and improve the outcome (*Spiekermann et al., 2018*).

The original NOSE was specifically developed for assessing nasal obstruction regardless of the cause (*Stewart et al., 2004*). It has been translated into many languages with good reliability and validity. In the present study, our evaluation of the Malay version of NOSE demonstrated that it has good validity, reliability and responsiveness to support its application as a tool for assessing patients with nasal obstruction. All participants involved were able to fully complete the questionnaire without any assistance in a short period of time. This signifies that they were able to understand the questions and comfortable to answer all of the items in the scale. This implies that the M-NOSE is acceptable to be used in an outpatient setting and not burdensome to patients.

A reliable measurement tool requires adequate test-retest reliability and good internal consistency. A test re-test reliability certifies the measurements attained at a specific time is both representative and stable over time and denotes the internal validity of the tool. In the original study a correlation coefficient of at least 0.7 was considered adequate in test-retest reliability for NOSE (*Stewart et al., 2004*). The M-NOSE has correlation coefficient more than 0.7 for all five items and total score demonstrating test re-test reliability analogous to the English and other adapted versions. The internal consistency describes the degree of relationship among the items in the test and checks to ensure all the items are measuring the concept they are supposed to be measuring. The M-NOSE has equivalent internal consistency as the English version with Cronbach's alpha of 0.81 compared to 0.78 of the English versions.

The inter-item correlation and correlation between each item and the total score of M-NOSE demonstrated moderate to strong correlation between all items and total score, and fair to moderate correlations of inter-items. These correlations are comparable to the English version and confirmed the good construct validity of the M-NOSE. The discriminatory validity is to determine the ability of a tool to detect the presence or absence of the disease. The comparison of the control with the intervention groups in the present study showed a highly significant difference between the two groups confirming that M-NOSE has a good discriminatory validity similar to the English version. In clinical practice, an assessment tool must be responsive and sensitive to any significant change in the patient's health to be beneficial. Even though the original NOSE has only five items, it has been shown to be sensitive to changes in nasal obstruction (*Stewart et al., 2004*). Likewise, the M-NOSE showed satisfactory sensitivity to change in the current study. The M-NOSE demonstrates good responsiveness with a significant reduction of mean symptom score postoperatively.

Understanding the differences in the characteristics and responses of two separate groups in questionnaire studies may offer useful insights into the impact of specific conditions. Healthy individuals who served as the control group had normal nasal airflow and did not have difficulty breathing through their nose. Their responses can aid in establishing normative values and determining what is considered normal in terms of symptoms, quality of life, or other metrics being evaluated. They are likely to report fewer nasal function issues and, overall, higher quality-of-life indicators related to nose health. Responses may be more consistent and less variable since they do not experience the changes associated with nasal obstruction. The second group with nasal obstruction, on the other hand, may face greater impacts on everyday activities and general health due to the chronic nature of their disease. Their questionnaire responses are likely to reflect these issues, with less favorable results in areas relating to nasal function. Responses may vary more widely due to variances in the severity of the obstruction and individual coping techniques. The impact of treatment on individuals with nasal obstruction revealed improvements in responses post-treatment, suggesting the potential benefits of corrective procedures for those with a deviated nasal septum. The questionnaire highlights the benefits of addressing nasal obstructions in impacted individuals and may aid in selecting appropriate candidates for specific treatment procedures.

The original English version has been shown to be valuable in determining the successful outcome of medical and surgical treatments in several studies. A study had used NOSE to assess the response of allergic rhinitis patients following treatment by intranasal corticosteroids and the improvement of symptoms was reproduced accurately by the NOSE (*Dutta et al., 2020*). Such quantification allows the exact evaluation of the response towards medical therapy by which unnecessary surgery could be avoided. On the other hand, surgeons need a reliable method to justify the indication of surgery and evaluate the outcome when surgery is being contemplated. The effectiveness of different surgical techniques to address nasal obstruction can be appraised reliably by the NOSE (*Stewart et al., 2004*; *Shukla, Nemade & Shinde, 2020*). The NOSE has been used to demonstrate that septoplasty was an effective procedure to relief nasal obstruction (*Kahveci et al., 2012*). Furthermore, septoplasty with partial inferior turbinectomy was shown to exhibit greater improvement symptomatically compared to septoplasty alone by using NOSE (*Dinesh Kumar & Rajashekar, 2016*). The responsiveness and stability of NOSE allow the surveillance of symptom change over time. The use of NOSE demonstrated significant reduction of nasal obstruction was achieved at 1 month following septoplasty and turbinate reduction with the status quo remains unchanged at 6 months (*Law et al., 2021*). A reliable postoperative evaluation is necessary as burden of disease varies from patient to patient and identification of problematic cases allows individualized management (*Gerecci et al., 2019*). As M-NOSE demonstrates similar psychometric properties like the original English version, corresponding benefits could be obtained with M-NOSE in delineating the response towards treatment, in addition to being an essential tool in health quality research. The assessment of the improvement of symptoms by a self-administered questionnaire without the confirmation by an objective method can be seen as the limitation of this study. However, this does not compromise the overall reliability and validity of the M-NOSE.

## CONCLUSIONS

The Malay version of NOSE has satisfactory reliability, internal consistency, reproducibility and responsiveness comparable to the English version. It is a valid and convenient tool in the assessment of the impact and treatment outcome of nasal obstruction. It can be applied in daily clinical practice and research amongst Malay speaking patients.

## ACKNOWLEDGEMENTS

The authors would like to thank Madam Nurul Fitriah Abd. Rashid and Madam Nurzulaikha Mahd Ab.Lah for the methodology/statistical assistances and Madam Fadilah Zakaria and Madam Nur Aisyah Md. Ridzuan for the linguistic assistance.

### Funding

The authors received no funding for this work.

### Competing Interests

The authors declare there are no competing interests.

### Author Contributions

- Addina Mohd Baki conceived and designed the experiments, performed the experiments, analyzed the data, prepared figures and/or tables, authored or reviewed drafts of the article, and approved the final draft.
- Suzina Sheikh Ab Hamid conceived and designed the experiments, performed the experiments, analyzed the data, authored or reviewed drafts of the article, and approved the final draft.
- Rosli Mohd Noor performed the experiments, authored or reviewed drafts of the article, and approved the final draft.
- Baharudin Abdullah conceived and designed the experiments, performed the experiments, analyzed the data, prepared figures and/or tables, authored or reviewed drafts of the article, and approved the final draft.

### Human Ethics

The following information was supplied relating to ethical approvals (i.e., approving body and any reference numbers):

Human Research Ethics Committee, Universiti Sains Malaysia (USM)

### Data Availability

The raw data can be found in Supplemental File.

### Supplemental Information

Supplemental information for this article can be found online at http://dx.doi.org/10.7717/peerj.17825#supplemental-information.

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
