# Peer review of "Translation and validation of the Bahasa Malaysia version of the Nasal Obstruction Symptom Evaluation scale (M-NOSE)"

_PeerJ, doi:10.7717/peerj.17825_

## Round 0.1 · original submission · Minor Revisions

Dear Dr. Abdullah,

Please submit a revised manuscript, including a response to each of the reviewers' comments, and also explaining how you have revised the manuscript in response to the reviewers' comments.

Yours,

Yoshi

Prof. Yoshinori Marunaka, M.D., Ph.D.
Academic Editor
PeerJ Life & Environment

Reviewer 1 ·

Basic reporting

No comment

Experimental design

No comment

Validity of the findings

No comment

Additional comments

This study is well written and commendable. The rigorous cross-cultural adaptation process, robust psychometric properties, and significant differences observed between patients and controls highlight the validity and reliability of M-NOSE. It will be a valuable tool for assessing nasal obstruction impact and treatment outcomes, especially in Malaysia.

·

Basic reporting

The article was clearly written and understandable with good professional language. The format was following the manuscript format. The literatures were sufficient to the content of the whole text.

Experimental design

The experimental design was clearly outlined and explained in a professional manner. There was mentioned regarding cross-cultural adaptation, however, I could not find the adaptation besides the usual translation.

Validity of the findings

It will be great if the author could explore more on the validity analysis of the study. Although only 5 items in M-NOSE, the usage of repetitive association between each different visit of same patients could be expanded to a more exploratory analysis in the case of cultural adaptation is needed.

Additional comments

This is a good and useful study so that M-NOSE can be used in future for rhinology research.

Reviewer 3 ·

Basic reporting

No comment.

Experimental design

No comment.

Validity of the findings

If one compares the F and M in the Gender Column of the raw data and the Table 1, there seems to be a discrepancy. Also, there seems to be a discrepancy between the SD of age of control group of the raw data and Table 1. Kindly check it.

Additional comments

I am satisfied with what I have read.
Some minor comments that need correction:
1. Abstract: Kindly change “nasal block” to “nasal blockage” or “nasal obstruction”.
2. Abstract: Kindly mention the study design in the methods. Was it a prospective questionnaire validation study?
3. Title: acceptable
4. Line 91: Kindly change the word “secondary” to “due”.
5. Line 94: Kindly explain how you excluded subjects with allergy. Did you perform any prick test, or blood IgE test, or did you ask them?
6. Line 96-97: Since there is actually no real validated objective method that defined “and normal finding on endoscopic examination”, I would simply replace this sentence with “who were no candidates for nasal surgical procedure”.
7. Line 117: was it really a 10 cm VAS or did you mean 10-points VAS?
8. Line 120: Similarly, was it really a 100 mm? Why not 10 cm?
9. Table 1: Kindly mention age unit (years)
10. Line 157: Kindly use past tense (had instead of has)
11. Table 2: Kindly mention the meaning of abbreviation IQR.
12. Line 161: Kindly write down “The correlations were” instead of “The correlation was”.
13. Line 177: Kindly correct the typo found and replace “The” with “There”.
14. Kindly include in the discussion a short paragraph describing the study population. Were subjects in the intervention group representative of subjects with nasal obstruction and were control subjects representative of subjects without nasal obstruction? (Yes, they were, just state this for the sake of good discussion).
15. Kindly include comments in your “raw data” file, which indicate what is every column.
16. If one compares the F and M in the Gender Column of the raw data and the Table 1, there seems to be a discrepancy. Also, there seems to be a discrepancy between the SD of age of control group of the raw data and Table 1. Kindly check it.

---

## Round 0.2 · accepted · Accept

Dear Dr. Abdullah,
Congratulations again, and thank you for your submission.
Warm regards,
Yoshi
Prof. Yoshinori Marunaka, M.D., Ph.D.

Reviewer 3 ·

Basic reporting

No comment.

Experimental design

No comment.

Validity of the findings

No comment.

Additional comments

No comment.